# EdgeCape: Edge Weight Prediction For Category-Agnostic Pose Estimation

**Or Hirschorn and Shai Avidan**
Tel Aviv University
`https://orhir.github.io/edge_cape`

## Abstract

Category-Agnostic Pose Estimation (CAPE) localizes keypoints across diverse object categories with a single model, using one or few annotated support images. Recent works have shown that using a pose-graph (i.e., treating keypoints as nodes in a graph rather than isolated points) helps handle occlusions and break symmetry. However, these methods assume a given pose-graph with equal-weight edges, leading to suboptimal results. We introduce EdgeCape, a novel framework that overcomes these limitations by predicting the graph's edge weights in order to optimize localization. To further leverage structural (i.e., graph) priors, we propose integrating Markov Attention Bias, which modulates the self-attention interaction between nodes based on the number of hops between them. We show that this improves the model's ability to capture global spatial dependencies. Evaluated on the MP-100 benchmark, which includes 100 categories and over 20K images, EdgeCape achieves state-of-the-art results in the 1-shot and 5-shot settings, significantly improving localization accuracy. Our code is publicly available.

## 1 Introduction

2D pose estimation, which involves identifying the locations of key semantic parts within an image, is a fundamental problem in computer vision. From human pose estimation (Fang et al., 2022a; Cao et al., 2019; Yang et al., 2021) to animal tracking and vehicle localization (Song et al., 2019; Reddy et al., 2018), accurate pose estimation is essential for both academic research and industrial applications. Traditional approaches mainly focused on category-specific models that are tailored to pre-defined keypoints and pre-defined object categories. These models achieve high accuracy in some domains but struggle when encountering categories that lack annotated training data. This limitation has sparked interest in flexible models that generalize beyond the fixed categories and keypoints seen in training.

To address these challenges, Category-Agnostic Pose Estimation (CAPE) has emerged as a promising solution (Xu et al., 2022a). CAPE enables keypoint localization for arbitrary keypoints and any object category using only a few annotated support images, allowing a single model to generalize across diverse object types. By significantly reducing the need for extensive data collection and retraining for each new category or keypoint definition, CAPE provides a versatile and cost-effective approach to pose estimation. However, CAPE remains exceptionally challenging as it must infer keypoint relations that are structurally meaningful across vastly different object categories using a few annotated examples. This is crucial for breaking symmetry, handling occlusions, and preserving object structure. Figure 1 highlights the differences between recent CAPE methods. Although early works treated keypoints as isolated entities, recent works (Hirschorn & Avidan, 2024; Rusanovsky et al., 2024) address this challenge by leveraging user-defined structural graphs. These graphs describe the skeletal relations between keypoints and are termed *pose-graphs*. These structural priors hold for any viewing angle and both for rigid and non-rigid objects. Graph-based methods provide robustness to occlusions, handle symmetric structures, and enforce anatomical consistency.

While effective, these methods rely on user-provided unweighted pose-graphs, significantly limiting their adaptability. For example, when locating a human's right elbow, both the right hand and shoulder provide useful context, but their contributions should vary in strength. In conventional pose estimation, these relative contributions are learned by leveraging the fact that the object category is

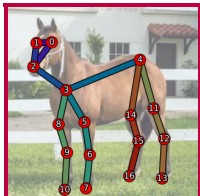
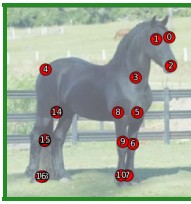
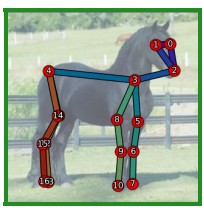
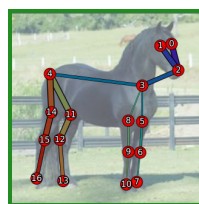

| Support Data | CapeFormer (Only Nodes) | GraphCape (Unweighted Graph) | **EdgeCape (Ours)** (Weighted Graph) |

Figure 1: Given a support image, keypoints definition, and skeletal relations (support data) from any category, our model localizes the keypoints on a query image. Previous methods treat keypoints as isolated (CapeFormer (Shi et al., 2023b)) or use unweighted graphs (GraphCape (Hirschorn & Avidan, 2024)). We, in contrast, predict weighted graphs that lead to better localization.

fixed, meaning its pose-graph definition is shared and can be learned from abundant same-category examples. In contrast, CAPE presents fundamentally new and largely unexplored challenges: object categories are unknown, and keypoints are determined dynamically at test time. We aim to advance graph-based CAPE by leveraging optimal pose-graphs with real-valued edge weights. However, multiple challenges arise. First, humans struggle to determine the optimal pose-graph, as demonstrated in Figure 2. This is crucial, as different graph definitions significantly impact localization performance (Hirschorn & Avidan, 2024). Moreover, requiring users to manually specify edge weights is impractical, as there is no clear correct assignment. Second, existing edge prediction methods are unsuitable for CAPE. Self-attention-based approaches, such as Graph Attention Transformers (Veličković et al., 2017), struggle in category-agnostic settings because they fail to model structural keypoint relations (Hirschorn & Avidan, 2024). Alternatively, GCN variants (Cai et al., 2019; Shi et al., 2019) assume a fixed pose-graph - an assumption that does not hold for CAPE. Lastly, CAPE requires generalization to unseen object categories, making direct pose-graph prediction particularly challenging, as it demands an implicit understanding of 3D object structure and anatomy. Thus, a more sophisticated approach is needed.

In this paper, we introduce *EdgeCape*, a novel approach that extends the graph-based CAPE framework by tackling the challenge of category-agnostic pose-graph prediction. Our goal is to find the optimal weighted pose-graph for any class. Rather than inferring the entire pose-graph from scratch, we refine a user-provided unweighted pose-graph by learning to assign edge weights and add or remove connections as needed. This approach eliminates the need for 3D prior knowledge and enables the model to adapt to any object category with varying geometries using a simple structural prior. Furthermore, our method learns subtle instance-specific adjustments, which are crucial to handling CAPE's inherent ambiguity. In addition, we overcome the lack of ground-truth optimal edge weights through a self-supervised strategy that predicts pose-graphs to maximize localization accuracy and improve occlusion handling. Finally, to better exploit pose-graphs for improved localization, we introduce a Markov attention bias that adjusts the model's self-attention according to the structural distance between nodes in the graph, supporting more complex spatial relations.

We evaluate our approach on the MP-100 benchmark, a comprehensive dataset comprising over 20,000 images spanning 100 diverse categories. Our method outperforms the previous state-of-the-art, significantly improving localization accuracy. We also demonstrate the robustness of our approach under challenging conditions through an extensive ablation study.

In summary, we introduce a novel CAPE method that enables a more nuanced capture of object geometry. Our key contributions are:

- We are the first to explore category-agnostic pose-graph prediction, introducing a novel mechanism that learns instance-specific weighted graphs that generalize across unseen object categories.
- We propose an enhanced graph-based CAPE architecture with Markov Attention Bias, allowing the model to better capture complex spatial dependencies between keypoints.
- We achieve SOTA results on the MP-100 benchmark, demonstrating the effectiveness of our approach in both 1-shot and 5-shot settings.

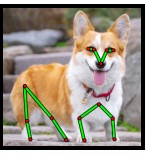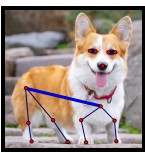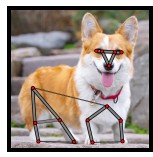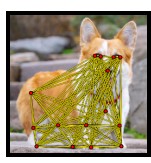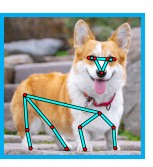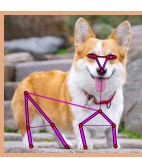

Figure 2: **Which of these pose-graphs is optimal?** As edge placement can be ambiguous, we aim to learn the optimal pose-graph for category-agnostic keypoint localization. We show several graph annotations for the same image. The second-to-last (cyan) serves as input to our model; the last (orange) is our predicted pose-graph. While not visually superior, it achieves the best performance.

## 2 RELATED WORKS

### 2.1 CATEGORY-AGNOSTIC POSE ESTIMATION

Category-agnostic pose estimation (CAPE), introduced by Xu(Xu et al., 2022a), aims to extend conventional category-specific pose estimation (Fang et al., 2022a; Cao et al., 2019; Yang et al., 2021; Yu et al., 2021; Yang et al., 2022) and multi-category pose estimation (Xu et al., 2022b; Yu et al., 2021) to unseen categories, enabling models that generalize beyond category-specific training for greater flexibility and robustness. POMNet (Xu et al., 2022a), a regression-based approach (Li et al., 2021; Oberweger & Lepetit, 2017; Zimmermann & Brox, 2017), utilized a transformer to encode query images and support keypoints for similarity prediction. Building on this, CapeFormer (Shi et al., 2023a) adopted a DETR-like framework (Carion et al., 2020; Zhang et al., 2022; Fang et al., 2022b; Wang et al., 2022) to address unreliable matching outcomes, refining initial predictions. Later, different approaches were suggested to improve various aspects of CapeFormer. ESCAPE(Nguyen et al., 2024), introduced super-keypoints that capture the statistics of semantically related keypoints from different categories, tackling variability in object appearances and poses. Chen (Chen et al., 2024) predicted meta-points independent of support annotations, later refined to align with target keypoints. Similar to object proposal tokens (Alexe et al., 2012; Hosang et al., 2015; Zitnick & Dollár, 2014), meta-points also provide structural cues. SCAPE (Liang et al., 2024) simplified the task by directly regressing keypoint locations without refinement. X-Pose (Yang et al., 2024) introduced a bottom-up approach for multi-instance localization. PPM (Peng et al., 2024) harnessed Stable Diffusion (Rombach et al., 2022), learning pseudo-prompts for each keypoint at test time and localizing via diffusion model cross-attention maps. This idea was later expanded by Chen et al. (2025b) which explored weak-shot keypoint estimation using intra-image and inter-image correspondences. FMMP (Chen et al., 2025a) employed multiscale features and leveraged denser keypoint localization by randomly mixing structurally linked keypoints. While effective, its reliance on multi-scale feature extraction restricts the range of backbone architectures that can be utilized. CapeLLM Kim et al. (2025) uses text in order to locate keypoints, using LLMs to replace the need for support examples. Recent work highlights the importance of object structure for CAPE localization. GraphCape (Hirschorn & Avidan, 2024) used graph convolutional networks (GCNs) (Kipf & Welling, 2016) to model structural relations between keypoints, improving robustness to symmetry and occlusions. CapeX (Rusanovsky et al., 2024) extended this by integrating pose-graphs with textual point explanations, enabling pose inference from natural language. SDPNet (Ren et al., 2024) also adopted graph-based methods, employing an auxiliary GCN for information sharing between keypoints. It predicts adjacency matrices via keypoint self-attention, supervised by a secondary GCN and a mask-reconstruction task for self-supervised learning. However, these methods face key limitations: (1) they rely only on keypoint features, whereas object structure may be better inferred from the entire object; (2) generalizing to unseen categories without prior knowledge is highly challenging and requires 3D anatomical understanding; and (3) structural cues may be encoded in the auxiliary GCN rather than the adjacency matrix, leading to information loss since the GCN is used only during training. In contrast, we combine image and keypoint features to refine prior structural knowledge and leverage decoder supervision to build a weighted graph optimized for localization. Although category-agnostic graph prediction remains underexplored, AutoLink He et al. (2022) validates the utility of learned weighted graphs for pose estimation. However, it is constrained to a category-specific setting and a fixed skeletal topology. In contrast, CAPE necessitates category-agnostic generalization, adapting to varying object categories and keypoint definitions at test time.

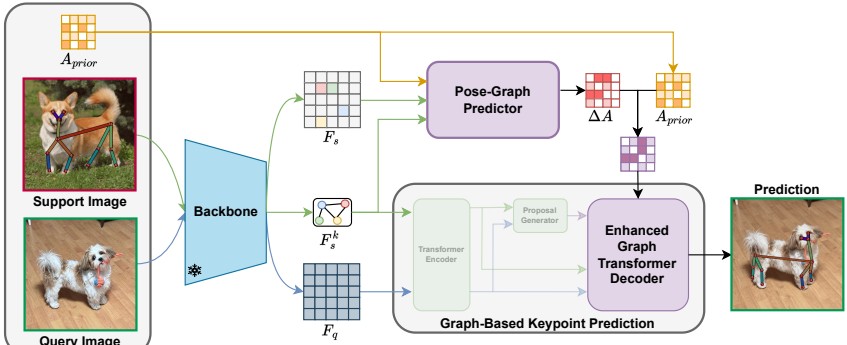

Figure 3: **Framework Overview.** Our model consists of two main components: a pose-graph predictor (visualized in Figure 15) and a graph-based keypoint predictor. The pose-graph predictor refines the prior graph input by predicting residual connections. The graph-based keypoint predictor then utilizes the predicted keypoint relations, improving localization across diverse object structures.

## 2.2 STRUCTURAL ENCODING IN GNNS

Graph Transformers extend Transformers to graph-structured data, where self-attention can be viewed as message passing between all nodes, independent of connectivity. Structural information is incorporated using three main strategies:

**Positional Embedding from Graph Structure.** Mialon and Feldman (Mialon et al., 2021; Feldman et al., 2022) introduce positional encodings based on graph kernels. Dwivedi (Dwivedi & Bresson, 2020) used Laplacian eigenvectors, later extended by Kreuzer (Kreuzer et al., 2021) with spectral encodings for greater expressivity. Ma (Ma et al., 2023b) instead modeled adjacency matrices as stochastic processes, using random-walk probabilities as relative encodings.

**Integrating Graph Neural Networks (GNNs) with Transformers.** Wu (Wu et al., 2021) captured local features with GNNs followed by Transformers to model long-range dependencies. Rampášek (Rampášek et al., 2022) combined GNN and self-attention layers, while Hirschorn (Hirschorn & Avidan, 2024) replaced Transformer feed-forward layers with GCNs.

**Incorporating Graph Structural Bias into Self-Attention.** Ying (Ying et al., 2021) encodes shortest-path distances as biases that directly alter the attention mechanism. Zhao (Zhao et al., 2021) incorporated proximity-based neighborhood relations. Dwivedi (Dwivedi & Bresson, 2020) integrated edge features into attention, while Wu *et al.* (Wu et al., 2022) injected topological information as relational biases to enhance attention fidelity.

## 3 METHOD

The full framework of our method is shown in Figure 3. In the following, we outline the key components of EdgeCape: (1) our graph-based CAPE formulation, (2) the category-agnostic pose-graph prediction network, and (3) an enhanced graph decoder that embeds structural biases. Additional details are in the supplementary (Section C).

## 3.1 PRELIMINARIES

CapeFormer (Shi et al., 2023b) treats keypoints as individual entities, sharing information via self-attention. While effective for single-category pose estimation, it struggles to generalize structural understanding to unseen categories (Hirschorn & Avidan, 2024). Consequently, self-attention alone is insufficient to capture structural relations for CAPE. GraphCape (Hirschorn & Avidan, 2024) addresses this limitation by incorporating user-provided pose-graphs as a structural prior. The pose-graph is represented by an adjacency matrix $A_{prior} \in \{0, 1\}^{K \times K}$, where $a_{ij} = 1$ if node $v_i$ is connected to node $v_j$, and 0 otherwise. Using graph convolutions, information is propagated between connected keypoints, aiding in structure preservation and occlusion handling. Additionally, integrating graphs during training helps break feature symmetry by enforcing similarity based on

structural relations rather than purely semantic attributes. Our work is built on GraphCape, and we include the full framework details in the supplementary (Section C.1).

## 3.2 CATEGORY-AGNOSTIC POSE-GRAPH PREDICTION

Category-agnostic pose-graph prediction presents a fundamentally new and largely unexplored challenge: object structures vary widely across categories, making it difficult to infer structure from scratch. Our key insight is to predict adjustments to a user-provided prior $A_{\mathrm{prior}}$ rather than learning the entire graph. This yields expressive, weighted pose-graphs that adapt to object-specific structures while remaining grounded in prior knowledge. Formally, we define a learnable function $f_\theta$ to learn the residual pose-graph in the form of an adjacency matrix $\Delta A \in \mathbb{R}^{K \times K}$:

$$\Delta A = f_\theta(A_{prior}, F_s, F_s^k) \tag{1}$$

where $A_{prior} \in \{0, 1\}^{K \times K}$ is the unweighted graph input provided by the user (like in GraphCape), and $F_s \in \mathbb{R}^{hw \times C}$ and $F_s^k \in \mathbb{R}^{K \times C}$ are the support image and keypoint features.

**Refining Structural Features.** First, we refine the keypoint features $F_s^k$ to enhance their structural semantics. Rather than sharing information only among the keypoint features $F_s^k$, we also incorporate the support image features $F_s$ to provide valuable global structural context, which is especially important in the category-agnostic setting where objects' orientations vary widely and keypoints alone may be ambiguous.

We build on a graph transformer decoder (Hirschorn & Avidan, 2024), which is well-suited for modeling structural dependencies. To strengthen the interaction between $F_s^k$ and $F_s$, we add a cross-attention layer where the image features are updated using the keypoint features. We refer to this as a dual-attention graph decoder, since it enables bidirectional exchange of structural information between image and keypoint features. In practice, we find that refining image features alongside keypoint features significantly improves the quality of learned structural representations. The architecture of our pose-graph prediction network is illustrated in the Supplementary (Figure 15).

This modification also changes the role of the decoder itself. Originally designed to exchange information between support keypoints and a query image, it now jointly processes keypoints and support image features of the same image. This repurposing allows the model to learn richer structural priors directly from the support image, leveraging the fact that structural patterns are shared within object categories.

**Predicting Residual Edges.** Using the refined structure-aware features $F_{refined}^k$, we compute $\Delta A$ as pairwise cosine-similarities:

$$\Delta A_{ij} = < F_{refined}^i, F_{refined}^j > \tag{2}$$

This similarity measure naturally quantifies the directional alignment and strength of relations between keypoints. We found this similarity measure as a balance between accuracy and efficiency. Alternatives such as MLPs or attention-based predictions yielded only marginal improvements at a higher computational cost.

The resulting $\Delta A$ is then combined with the user-provided prior $A_{\mathrm{prior}}$. A naïve approach would directly sum $A_{\mathrm{prior}}$ and $\Delta A$, but this leads to unstable training and poor convergence, as the output graphs at the start of training lack structural meaning. Instead, we introduce a simple yet effective scaling mechanism that stabilizes training. We combine $A_{\mathrm{prior}}$ and $\Delta A$ using a learnable scaling factor initialized to zero, and pass the result through an activation function to ensure positivity:

$$A' = \mathrm{ReLU}(A_{\mathrm{prior}} + c\Delta A) \tag{3}$$

where $c$ is the learnable scalar. At the start of training, $c = 0$, meaning the output pose graph is initially $A_{\mathrm{prior}}$, providing a stable foundation.

Lastly, we enforce adjacency matrix symmetry and normalization to ensure proper interpretation as a valid pose-graph, and right-stochasticity. Symmetry is enforced by averaging the matrix with its transpose and normalization is applied row-wise:

$$A = \frac{A' + A'^T}{2} \tag{4} \qquad\qquad \tilde{A}_{ij} = \frac{A_{ij}}{\sum_j A_{ij}} \tag{5}$$

**Pose-Graph Prediction Self-Supervision.** Training the pose-graph predictor with only a localization loss does not reliably produce useful graphs. We therefore introduce an additional supervision signal. Designing such a signal is challenging: even for humans, identifying the correct graph structure is non-trivial, and assigning precise edge weights is impractical. To address this, we adopt an unsupervised masking strategy, where our keypoint prediction module provides indirect supervision for the pose-graph predictor. Using masking, we aim to find adjacency matrices that can help the model overcome occlusions, a realistic scenario for this task.

Optimization is done as follows. We randomly mask a subset of the input support keypoint features using the vector mask $M$, and replace them with a learnable masking token $F_{mask}$. Then, using the masked input, the model predicts the query keypoint locations:

$$P_i^m = g_\theta(M \circ F_s^k + (1 - M) \circ F_{mask}, \tilde{A}, P^0) \tag{6}$$

where $\circ$ is the Hadamard product, $g_\theta$ is the keypoints prediction module, $P^0$ are the predicted coordinates from the proposal generator (Section C), and $\tilde{A}$ is the predicted adjacency matrix. The adjacency loss is the $L_1$ distance between the predicted locations $P^m$ using masked keypoint features and the ground-truth locations $\hat{P}$:

$$\mathcal{L}_{adj} = \sum_{i=1}^{K} |P_i^m - \hat{P}_i| \tag{7}$$

During backpropagation, this loss is used to update only the adjacency predictor. To achieve this, we freeze the decoder's weights and all inputs except $\tilde{A}$, ensuring that the adjacency matrix itself encodes information useful for overcoming occlusion.

The final training objective combines this adjacency loss with the standard localization loss:

$$\mathcal{L} = \mathcal{L}_{offset} + \lambda_{adj}\mathcal{L}_{adj}, \tag{8}$$

where $\mathcal{L}_{offset}$ is $L_1$ localization loss as in (Shi et al., 2023b). More details about our training scheme are in the supplementary, Section C.4.

### 3.3 MARKOV ATTENTION BIAS

In CAPE, encoding spatial relations between keypoints is critical for robust localization (Hirschorn & Avidan, 2024; Rusanovsky et al., 2024). Having established a way to predict more informative adjacency matrices, we now focus on how to better leverage these spatial relations during localization. Keypoint localization is done using a transformer decoder (Figure 3) where the self-attention mechanism provides a global receptive field by considering all pairwise keypoint interactions. However, this flexibility comes at a cost, as it weakens the influence of structural relations. We aim to improve the model's ability to capture both local and long-range structural dependencies. Graph-Cape (Hirschorn & Avidan, 2024) partially tackled this issue by integrating GCNs into the feed-forward network. While effective at modeling local structure, GCNs still struggle with reasoning over long-range relations and therefore do not provide a complete solution. To address this limitation, we introduce a new graph-informed bias term (Ying et al., 2021) into the self-attention mechanism, enabling structural connectivity patterns to directly influence the flow of information and enhance overall performance.

Our predicted adjacency matrix $\widetilde{A}$ contains weighted edges, reflecting varying strengths of keypoint relations. We normalize it row-wise, treating it as a stochastic matrix that defines a Markov process. Under this view, $(\widetilde{A}^k)_{ij}$ represents the probability of reaching node $v_j$ from node $v_i$ in exactly $k$-hops. $\widetilde{A}_{ij}$ captures direct (1-hop) links while $(\widetilde{A}^k)_{ij}$ captures multi-hop (long-range) relations, providing a principled way to encode both local and global structure. We start by constructing the matrix $P$:

$$P_{ij} = [I, \widetilde{A}, \widetilde{A}^2, \ldots, \widetilde{A}^{K-1}]_{i,j} \in \mathbb{R}^K \tag{9}$$

where $K$ is the maximum number of hops considered. Intuitively, $P_{ij}$ captures multi-hop structural relations. Then, using an MLP: $\mathbb{R}^K \to \mathbb{R}$, we modulate the influence of keypoints based on graph distance, allowing the model to learn which relations are most relevant for localization. The resulting modified self-attention in the decoder is:

$$a_{ij} = \frac{(h_i W_Q)(h_j W_K)^T}{\sqrt{d}} + \text{MLP}(P_{ij}) \tag{10}$$

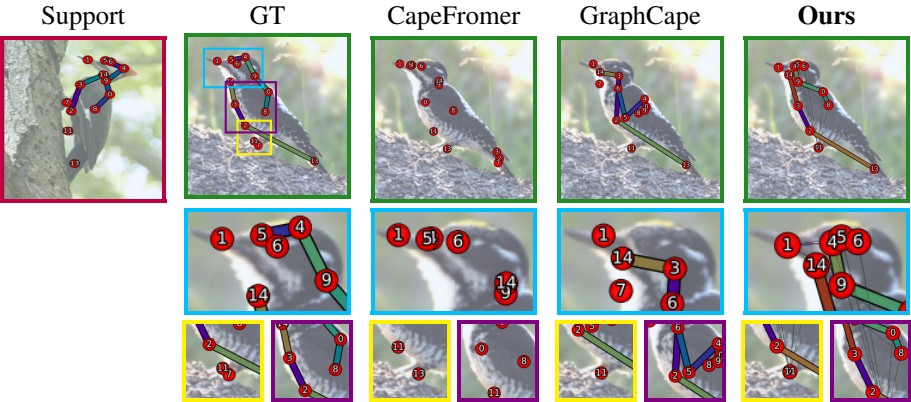

Figure 4: **Qualitative Comparison.** We visualize keypoint predictions for the 1-shot setting. The left column shows the support data, followed by ground-truth query keypoints, and results of different methods. Our method performs best by leveraging predicted weighted pose-graphs, which serve as more effective structural priors for keypoint localization.

This formulation preserves the expressiveness of self-attention while explicitly guiding it to respect structural priors. In effect, the model learns to weigh keypoint interactions based on both semantic similarity and graph connectivity. A figure of the new graph-transformer decoder, along with the full derivation of the bias term and training scheme, are provided in the supplementary, Section C.3.

## 4 EXPERIMENTS

We follow standard CAPE practice by training and evaluating on MP-100 dataset (Xu et al., 2022a), which includes over 20,000 images from 100 categories with up to 68 keypoints per category. The dataset is split such that categories used for training, validation, and testing do not overlap, and it provides five different category splits to ensure robust generalization. We also report results on AP-10K (Yu et al., 2021) (supplementary, Section B.2), an animal pose dataset covering 23 families and 60 species. Unlike MP-100, it uses a shared keypoint definition and pose-graph structure.

For evaluation, we use the Probability of Correct Keypoint (PCK) (Yang & Ramanan, 2012) metric. We use a 0.2 threshold for backward compatibility with prior works. But, as this metric became saturated, we follow (Chen et al., 2025a) and also report mPCK (mean PCK over thresholds 0.05, 0.1, 0.15, 0.2) for a more challenging evaluation.

**Implementation Details.** For a fair comparison, we match the training parameters, data augmentations, and data pre-processing to previous works. Additional details are in the supplementary (Section C.5). Notably, our method introduces only minimal latency overhead, adding a lightweight MLP (Markovian Attention Bias) and a single decoder pass (pose-graph predictor), adding only ∼2 ms per image (on an A5000 GPU). A detailed analysis of computation and complexity is in the supplementary.

### 4.1 QUALITATIVE RESULTS

Figure 4 provides a qualitative comparison between our method, CapeFormer (Shi et al., 2023b), and previous graph-based CAPE work GraphCape (Hirschorn & Avidan, 2024). CapeFormer does not use any graph prior, while GraphCape uses pre-defined graph prior. With stronger pose-graphs and a better mechanism to exploit them via Markov Attention Bias, our approach achieves superior localization.

Additionally, Figure 5 provides examples of category-agnostic pose-graph predictions. As can be seen, our network weakens symmetric parts' connections, which can hurt localization. For example, the table has stronger connections between the base and legs than between the different corners

Table 1: **MP-100 Results.** We compare our method against baselines without graph priors or using fixed graphs. Our method consistently outperforms others across all settings and data splits.

| Method | Graph Prior | 1-Shot | | | | | | 5-Shot | | | | | |
|---|---|---|---|---|---|---|---|---|---|---|---|---|---|
| | | Split 1 | Split 2 | Split 3 | Split 4 | Split 5 | Avg | Split 1 | Split 2 | Split 3 | Split 4 | Split 5 | Avg |
| CapeFormer | None | 78.44 | 73.56 | 73.61 | 73.14 | 73.51 | 74.45 | 83.71 | 79.77 | 79.18 | 79.62 | 75.89 | 79.63 |
| GraphCape | Fixed | 79.87 | 75.06 | 76.16 | 73.58 | 73.98 | 75.73 | 83.93 | 79.81 | 78.78 | 79.02 | 79.28 | 80.16 |
| **Ours** | **Predicted** | **81.96** | **77.63** | **77.35** | **75.68** | **75.98** | **77.72** | **84.92** | **80.91** | **79.68** | **79.76** | **80.08** | **81.07** |

Table 2: **Point-based Results.** PCK@0.2 performance comparison. Our approach outperforms other methods on the 1-shot and 5-shot settings. Best results are **bold**, second-best are underlined.

| Model | 1-Shot | | | | | | 5-Shot | | | | | |
|---|---|---|---|---|---|---|---|---|---|---|---|---|
| | Split 1 | Split 2 | Split 3 | Split 4 | Split 5 | Avg | Split 1 | Split 2 | Split 3 | Split 4 | Split 5 | Avg |
| POMNet (Xu et al., 2022a) | 84.23 | 78.25 | 78.17 | 78.68 | 79.17 | 79.70 | 84.72 | 79.61 | 78.00 | 80.38 | 80.85 | 80.71 |
| ESCAPE (Nguyen et al., 2024) | 86.89 | 82.55 | 81.25 | 81.72 | 81.32 | 82.74 | 91.41 | 87.43 | 85.33 | 87.27 | 86.76 | 87.63 |
| MetaPoint+ (Chen et al., 2024) | 90.43 | 85.59 | 84.52 | 84.34 | 85.96 | 86.17 | 92.58 | 89.63 | 89.98 | 88.70 | 89.20 | 90.02 |
| SDPNet (Ren et al., 2024) | 91.54 | 86.72 | 85.49 | 85.77 | 87.26 | 87.36 | 93.68 | 90.23 | 89.67 | 89.08 | 89.46 | 90.42 |
| X-Pose (Yang et al., 2024) | 89.07 | 85.05 | 85.26 | 85.52 | 85.79 | 86.14 | - | - | - | - | - | - |
| SCAPE (Liang et al., 2024) | 91.47 | 86.29 | 87.23 | 87.07 | 86.94 | 87.80 | 94.33 | 90.53 | **91.49** | **90.68** | 89.80 | 91.37 |
| FMMP (Chen et al., 2025a) | 88.19 | 85.26 | 83.03 | 85.29 | 84.72 | 85.30 | 91.97 | 89.36 | 87.35 | 89.33 | 88.56 | 89.31 |
| CapeFormer (Shi et al., 2023a) | 91.07 | 86.94 | 87.05 | 85.53 | 85.77 | 87.27 | 94.98 | 91.47 | 90.69 | 90.24 | 88.62 | 91.20 |
| GraphCape (Hirschorn & Avidan, 2024) | 92.39 | 88.47 | 89.24 | 85.76 | 86.66 | 88.50 | 95.21 | 91.51 | 90.65 | 89.92 | 90.22 | 91.50 |
| **Ours** | **93.57** | **89.53** | **89.31** | **87.38** | **87.29** | **89.42** | **95.45** | **91.88** | 91.13 | 90.28 | **90.61** | **91.87** |

of the base. Alternatively, it creates new helpful connections in the human face. In the supplementary (Section B.3), we demonstrate the instance adaptability of our graph prediction module.

## 4.2 QUANTITATIVE RESULTS

Following (Chen et al., 2025a), we categorize CAPE methods into two groups: graph and point-based methods. We note that point-based methods largely pursue orthogonal directions and can be combined. Additional metrics and in-depth comparisons are in the supplementary (Section B.1).

**Graph-based methods.** Table 1 reports mPCK results of graph-based methods on MP-100 under 1-shot and 5-shot settings. CapeFormer serves as a baseline, not using any prior, while GraphCape uses fixed pose-graphs. As shown, our method consistently outperforms other methods across all splits and settings. We achieve significant gains of +1.99% in 1-shot and +0.9% in 5-shot mPCK over GraphCape. At a finer evaluation threshold (e.g., 0.05), these margins widen substantially, reaching +4.0% and +2.23% respectively. Importantly, the improvements are stable across splits - even in cases where GraphCape underperforms CapeFormer (for example, split 3 in 5-shot), our approach remains superior. This suggests that simply injecting graph priors can be brittle, whereas predicting them yields a stronger and more reliable structural prior. Overall, we reach a substantial improvement of 3.27% (1-shot) and 1.44% (5-shot) mPCK over CapeFormer.

Input Graph      Predicted Graph

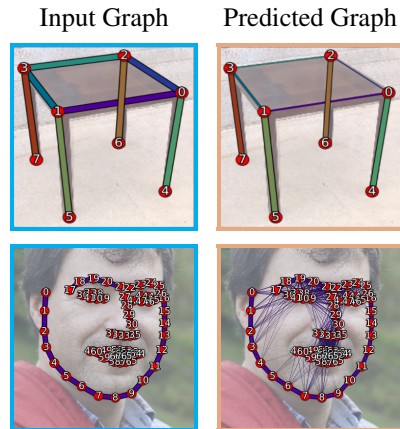

Figure 5: **Predicted Pose-Graphs.** We visualize predicted graphs: left column shows input $A_{\text{prior}}$, right shows output $A$. Line width reflects edge weight. Observe the slimmer table-base edges and the new facial edges. Our model prunes symmetric part links and forms connections that aid localization.

**Point-based methods.** We show in Table 2 our results compared to recent CAPE models using the more common, but saturated, PCK threshold 0.2. EdgeCape surpasses the previous state-of-the-art, with margins that are on par with - and in many cases exceeds - the gains reported by prior leading methods. As shown, at PCK@0.2 the improvements over GraphCape are +0.92% (1-shot) and +0.37% (5-shot), highlighting that performance using this coarse threshold is approaching saturation. Nonetheless, our method achieves impressive gains of +1.62% (1-shot) and +0.5% (5-shot) over SCAPE, the previous non-graph-based state-of-the-art. Notably, our work can be further combined with other point-based CAPE methods, which we leave for future work.

**Limitations.** Our method consistently enhances accuracy across both low- and higher-shot settings, with particularly notable improvements in the 1-shot scenario. Yet, the performance gain is smaller in higher-shot settings - a trend also observed when comparing CapeFormer and GraphCape. This is expected, as graph-based methods are particularly effective for handling occlusions and preserving structure under low-shot conditions. Nonetheless, our contribution in higher-shot settings remains substantial, as our learned pose-graphs and their effective utilization enable consistently stronger gains than those achieved by GraphCape.

### 4.3 ABLATION STUDY

Following standard practice, ablation studies are conducted on MP-100's split-1 in the 1-shot setting. We analyze the contribution of each component within our framework and examine the impact of input pose-graphs $A_{prior}$ on performance. In the supplementary (Section B.2), we provide additional extensive ablation experiments, including a quantitative comparison in the challenging super cross-category setting, performance using different backbones, evaluation without $A_{prior}$, evaluation of occlusion handling, quantitative analysis of different design choices and our suggested supervision strategy, and histograms of predicted adjacency weight changes.

Table 3: **Method Ablation.** We show mPCK results demonstrating the contribution of our category-agnostic pose-graph prediction module and the Markov Attention Bias.

|  |  | Markov Attention Bias | |
|---|---|---|---|
|  |  | (-) | (+) |
| **Pose-Graph** | (-) | 79.87 | 80.78 |
| **Prediction** | (+) | 80.76 | **81.96** |

**Components Contribution.** Table 3 shows the importance of both the Markov Attention Bias and the category-agnostic pose-graph prediction model. We build on GraphCape (Hirschorn & Avidan, 2024), incorporating each component separately. The results show that both components contribute to improved performance, with the greatest gain achieved when they are combined. The Markov Attention Bias not only improves performance on its own but also benefits further from the enhanced predicted pose-graphs, resulting in an even greater overall boost.

**The Impact of Prior Graph Input** $A_{prior}$**.** To evaluate the impact of the input pose-graph prior, we compare our method to GraphCape using noisy graph inputs $A_{prior}$, as shown in Figure 6. We inject up to 16 random edges per pose-graph, comparable to the original skeleton's edge count, spanning noise levels from mild to fully randomized topologies. Since graph-based models rely on connectivity for localization, incorrect connections mislead the model by propagating cues between unrelated keypoints, degrading performance. Yet, our method consistently outperforms GraphCape, with the gap widening as more random edges are added. This highlights our model's resilience to input pose-graph variations and its ability to predict optimal pose-

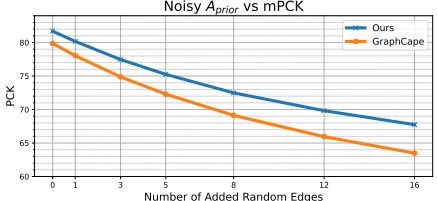

Figure 6: **mPCK with noisy $A_{prior}$.** Testing on graphs with randomly added edges shows that our method is robust to incorrect graph inputs, effectively predicting pose-graphs for improved localization.

graphs. Moreover, since the Markov Attention Bias heavily depends on the pose-graph, our superior performance further underscores the effectiveness of our prediction module. This robustness is especially valuable in real-world scenarios where defining an optimal pose-graph is difficult.

### 5 CONCLUSIONS

We introduce EdgeCape, a CAPE method that improves keypoint localization via predicted category-agnostic pose-graphs. Unlike previous approaches that treat keypoints independently or rely on unweighted user-provided graphs, EdgeCape predicts weighted pose-graphs for more accurate localization. By integrating these predictions with Markov Attention Bias, our model captures complex structural dependencies, improving robustness to occlusions and symmetry. On MP-100, EdgeCape achieves state-of-the-art performance in 1-shot and 5-shot settings. Beyond CAPE, our framework highlights the broader potential of weighted graph refinement for category-agnostic vision tasks under data scarcity and structural variability.

**Acknowledgment:** Part of this research was supported by ISF grant 2132/23.

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

## A  SUPPLEMENTARY - INTRODUCTION

Section B presents additional experimental results for our method, including:

- Additional MP-100 quantitative evaluations.
- Quantitative evaluations on the AP-10K dataset.
- Super-cross category experiment, emphasizing generalization.
- Occlusion handling experiment, using query image masking performance comparison.
- Supervision strategy ablation studies.
- Predicted adjacency matrix change - histogram showing the matrices change given different $A_{prior}$ inputs, demonstrating the effect of our graph prediction module.
- Additional qualitative results - including predicted graphs, and qualitative comparisons.

Section C provides additional details about the different components in our method:

- Framework Overview.
- Markov Attention Bias.
- Training Scheme.
- Implementation Details.

## B  FURTHER EXPERIMENTS

Table 4: **MP-100 PCK@0.1 Results.** comparison between graph-based methods on the MP-100 dataset. Our method consistently outperforms others across all settings and data splits.

| Method | 1-Shot | | | | | | 5-Shot | | | | | |
|--------|--------|--------|--------|--------|--------|--------|--------|--------|--------|--------|--------|--------|
| | Split 1 | Split 2 | Split 3 | Split 4 | Split 5 | Avg | Split 1 | Split 2 | Split 3 | Split 4 | Split 5 | Avg |
| CapeFormer | 74.48 | 68.39 | 67.99 | 68.18 | 68.86 | 69.58 | 81.46 | 77.12 | 76.28 | 76.71 | 71.86 | 76.69 |
| GraphCape | 76.17 | 70.05 | 71.15 | 68.89 | 68.95 | 71.04 | 81.99 | 77.22 | 75.26 | 75.99 | 76.28 | 77.35 |
| **Ours** | **79.29** | **73.82** | **73.1** | **71.16** | **72.19** | **73.91** | **83.32** | **78.42** | **76.29** | **76.82** | **77.29** | **78.43** |

Table 5: **MP-100 Additional Metrics.** AUC and NME performance under a 1-shot setting. Our approach outperforms other methods. The best results are **bold**.

| Model | AUC ↑ | | | | | | NME ↓ | | | | | |
|-------|-------|--------|--------|--------|--------|--------|-------|--------|--------|--------|--------|--------|
| | Split 1 | Split 2 | Split 3 | Split 4 | Split 5 | Avg | Split 1 | Split 2 | Split 3 | Split 4 | Split 5 | Avg |
| CapeFormer | 88.64 | 86.39 | 86.18 | 85.81 | 86.51 | 86.70 | 0.088 | 0.110 | 0.111 | 0.116 | 0.108 | 0.106 |
| GraphCape | 89.08 | 87.69 | 86.97 | 87.01 | 86.67 | 87.48 | 0.083 | 0.097 | 0.103 | 0.104 | 0.107 | 0.099 |
| **Ours** | **90.05** | **87.95** | **88.43** | **87.18** | **87.40** | **88.20** | **0.074** | **0.094** | **0.089** | **0.103** | **0.103** | **0.093** |

### B.1  QUANTITATIVE RESULTS - ADDITIONAL METRICS

**MP-100**    For a more complete comparison of our method, we report PCK at threshold 0.1 in Table 4. As shown in Table 4, our method outperforms GraphCape, former SOTA graph-based method by +2.87% on the 1-shot setting and by +1.08% on the 5-shot setting. Overall, we achieve gains of +4.33% and +1.74% over CapeFormer, a method that don't use any prior graph structure, emphasizing the importance of this added refined prior. Notably, even in splits where GraphCape underperforms CapeFormer, our method comes on top, underscoring the advantage of graph-based methods over point-based ones when optimal pose-graphs are learned and effectively utilized.

Additionaly we include the Area Under ROC Curve (AUC), and Normalized Mean Error (NME) on the MP-100 dataset 1-shot setting. As shown in Table 5, our method also excels in these metrics, demonstrating the superiority of our method.

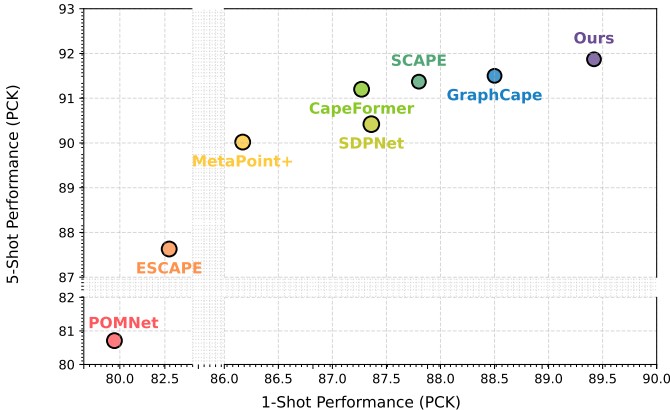

Figure 7: **MP-100 Comparison.** Average PCK@0.2 comparison with recent methods. The circle size indicates the model's size. The x-axis and y-axis represent the 1-shot and 5-shot PCK, respectively (best is top right). Our approach leads in both settings.

**Point-based methods**   We also include a visual quantitative comparison of the MP-100 dataset with recent CAPE methods in Figure 7. It's important to note that most other non graph-based CAPE works are complementary and parallel to ours, meaning that combining different methods could help build even stronger CAPE models.

**Computational Complexity.**   EdgeCape introduces a lightweight computational overhead, adding approximately 2.5M parameters and 2 GFLOPs to the GrapCape baseline (totaling $\approx 30$ GFLOPs). By leveraging recent transformer optimizations on modern hardware, this translates to a modest inference latency increase of only $\approx 2$ ms per image on an NVIDIA A5000. Notably, the matrix-power operations required for attention bias are executed using PyTorch's GPU-optimized linear algebra kernels, incurring negligible cost (measured at $\approx 0.02$ GFLOPs for 100 keypoints and $K = 4$). While we acknowledge that current CAPE methodologies have not yet reached real-time frame rates, we plan to investigate further efficiency improvements in future work.

## B.2   ABLATION STUDY

Table 6: **Backbones Ablation.** Comparison between graph-based methods on the MP-100 dataset (1-shot, split 1) using different backbones and metrics. Predicted weighted graphs improve performance, increasing the gap in performance using stronger pre-trained features.

| Method | Backbone | Metric | |
|---|---|---|---|
| | | PCK@0.1 | mPCK |
| CapeFormer | | 74.05 | 77.77 |
| GraphCape | SwinV2 | 76.44 | 79.80 |
| **Ours** | | **77.66** | **80.63** |
| CapeFormer | | 74.48 | 78.44 |
| GraphCape | DinoV2 | 76.17 | 79.87 |
| **Ours** | | **79.29** | **81.96** |

**Performance Using Different Backbones.**   Table 6 shows results on the MP-100 dataset under 1-shot setting. As can be seen, we consistently outperform CapeFormer (Shi et al., 2023b) and GraphCape (Hirschorn & Avidan, 2024), highlighting the effectiveness of our approach. Moreover, contrary to GraphCape's claim (Hirschorn & Avidan, 2024), DinoV2 features outperform SwinV2

in graph-based CAPE methods. While previous works (Lin et al., 2023; Shi et al., 2023b; Liang et al., 2024) have found other backbones more suitable for localization tasks, our results suggest that DinoV2's superior generalization and semantic understanding make it especially well-suited for category-agnostic settings. Importantly, we achieve these results while keeping the DinoV2 backbone frozen, preserving its rich self-supervised representations rather than fine-tuning, as done in prior methods. Furthermore, the performance gap increases when predicting pose-graphs using DinoV2 features, aligning with findings by Banani *et al.* (El Banani et al., 2024), who demonstrated the intrinsic 3D structural awareness of DinoV2. This structural understanding is particularly beneficial for capturing pose-graph structures.

Table 7: **AP-10K Results.** PCK@0.1, AUC, and NME results for the single-category AP-10K benchmark.

| Model | Metric | | |
|---|---|---|---|
| | PCK@0.1 $\uparrow$ | AUC $\uparrow$ | NME $\downarrow$ |
| CapeFormer | 69.70 | 86.41 | 0.094 |
| GraphCape | 72.01 | 87.22 | 0.086 |
| **Ours** | **72.48** | **87.29** | **0.085** |

**Single-Category Setting.** We report PCK performance on a single-category pose estimation benchmark using the AP-10K dataset (Yu et al., 2021), a large-scale animal pose estimation benchmark comprising 23 animal families and 60 species. This dataset features a unified keypoint definition and skeletal structure across all categories. Notably, the training and testing category splits are not mutually exclusive, meaning some categories may appear in both sets. This setup allows us to evaluate our method's effectiveness in a single-category context, where categories share common anatomical structures but exhibit species-specific variations.

Results are presented in Table 7. Our method outperforms CapeFormer (Shi et al., 2023b) and GraphCape (Hirschorn & Avidan, 2024), showing superiority also in this setting.

Table 8: **Super Cross-Category.** PCK@0.2 results for the super cross-category setting. Our method outperforms others in this challenging setting across most splits, demonstrating robust generalization.

| Method | Human Body | Human Face | Vehicle | Furniture |
|---|---|---|---|---|
| POMNet (Xu et al., 2022a) | 73.82 | 79.63 | 34.92 | 47.27 |
| ESCAPE (Nguyen et al., 2024) | 80.60 | 84.13 | 41.39 | **55.49** |
| MetaPoint+ (Chen et al., 2024) | 84.32 | 82.21 | **46.51** | 53.67 |
| SDPNet (Ren et al., 2024) | 83.84 | 81.24 | 45.53 | 53.08 |
| SCAPE (Liang et al., 2024) | 84.24 | 85.98 | 45.61 | 54.13 |
| CapeFormer (Shi et al., 2023a) | 83.44 | 80.96 | 45.40 | 52.49 |
| GraphCape (Hirschorn & Avidan, 2024) | 88.38 | 83.28 | 44.06 | 45.56 |
| **Ours** | **91.38** | **87.61** | 44.12 | 54.03 |

**Super Cross-Category.** We conduct a cross-super-category experiment following prior works to evaluate our model's generalization capacity. Splitting the MP-100 dataset into eight super-categories, we hold out each super-category in turn and train on the remaining categories. As shown in Table 8, our model outperforms other graph-based methods across most splits. Notably, in the furniture split, GraphCape (Hirschorn & Avidan, 2024) performs worse than CapeFormer (Shi et al., 2023b), indicating that in this case, the input pose-graph impedes performance. However, our graph-prediction network not only surpasses GraphCape but also outperforms CapeFormer. This demonstrates that our model successfully learns pose-graphs that improve localization, even when the input pose-graphs are suboptimal.

**Performance Without Prior Knowledge**    To assess the model's dependency on informative structural priors, we conducted an ablation test using fully-connected graphs, which essentially removes topology-based information. Under this setting GraphCape suffer a performance collapse, reverting to the baseline CapeFormer level Hirschorn & Avidan (2024). In contrast, EdgeCape demonstrates significant robustness, achieving a +1.29% mPCK improvement over CapeFormer even with uninformative fully-connected graphs. Notably, this performance is on par with what GraphCape achieves using its standard input prior structure. This confirms that our method provides distinct benefits even when the graph prior carries no useful information.

**Adjacency Matrix Supervision.**    To evaluate the effectiveness of our unsupervised masking strategy, we first test the performance using the GCN reconstruction approach proposed by Ren *et al.*(Ren et al., 2024), which utilizes an auxiliary GCN to reconstruct masked inputs. With this approach, we achieve (PCK@0.2) **92.88%** accuracy, compared to **93.57%** using our decoder-based reconstruction strategy - a decrease of **0.69%**. This drop supports our hypothesis that structural information is embedded within the auxiliary GCN weights and thus not retained during inference.

Furthermore, we investigate the impact of our decoder reconstruction strategy under varying levels of input keypoint masking. The results of this analysis are illustrated in Figure 8. As shown, masking keypoints enhances the supervision signal for the adjacency matrix, leading to more robust and structurally meaningful graph representations.

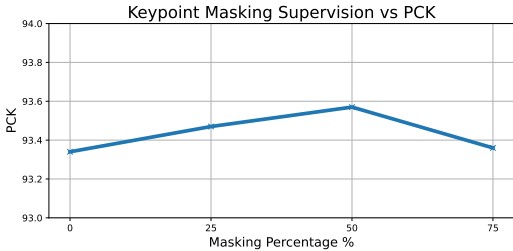

Figure 8: **Keypoint Masking Supervision.** Effect of the proposed unsupervised masking strategy with varying percentages of masked input keypoints

**Pose-Graph Predictor Design.**    We also ablate different design choices for the pose-graph predictor. First, we check the contribution of using full-image features compared to just the keypoint features to predict pose graphs. As hypothesized, using only keypoint features results in a decrease of **0.24%** compared to using all features. This validates that the full-image features' global context helps infer structural relations more effectively.

Moreover, we ablate the strategy of predicting pose graphs using the refined features. We compare the following strategies:

- Using the suggested cosine similarity:
$$\Delta A_{ij} = < F^i_{refined}, F^j_{refined} > \tag{11}$$

- Using an MLP-based prediction:
$$\Delta A_{ij} = \text{MLP}\left(\left[F^i_{\text{refined}} \parallel F^j_{\text{refined}}\right]\right) \tag{12}$$

- Using multi-headed attention-based prediction:
$$\Delta A_{ij} = \text{Linear}\left(\left[Q^i_1 \cdot K^j_1 \parallel \ldots \parallel Q^i_H \cdot K^j_H\right]\right) \tag{13}$$

All methods achieve similar performance. Thus, we opt for the simplest design and use cosine similarity as our preferred strategy.

**Handling Occlusions.**    To demonstrate the effect of weighted graph information in handling occlusions, we applied random partial masking to the query images before executing our algorithm. As shown in Figure 9, our method consistently surpasses others when parts of the query images are masked, accurately predicting keypoints.

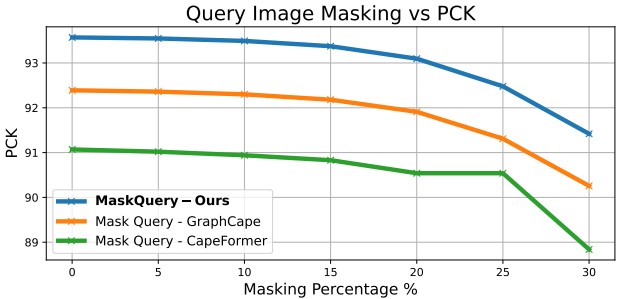

Figure 9: **Handling Occlusions.** PCK@0.2 results when masking the query image. Our method consistently surpasses GraphCape, leveraging cues provided by the weighted graph structure to overcome information gaps in the query images.

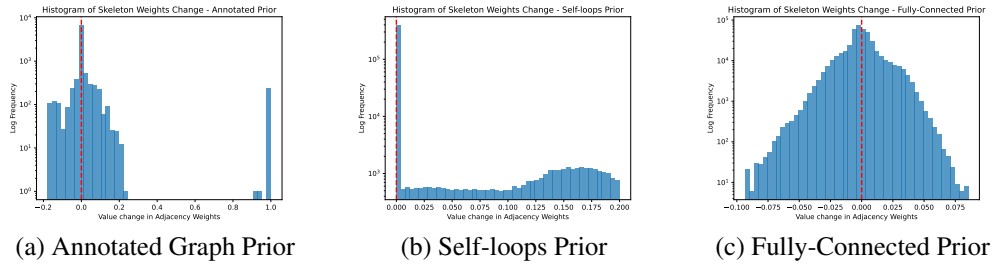

(a) Annotated Graph Prior        (b) Self-loops Prior        (c) Fully-Connected Prior

Figure 10: **Adjacency Matrix Change Histogram.** A histogram showing the weight changes of our predicted weighted graph. We show the change given different $A_{prior}$ inputs. the x-axis shows the change and the y-axis is the log-frequency. The dashed red line shows no change in edge weight.

**Adjacency Matrix Change.**   We assess how the weights of the adjacency matrices change. We create a histogram of the difference between normalized $A_{prior}$ and the normalized predicted $A$ matrix. Results are shown in Figure 10. demonstrate that our network effectively refines the input graph prior. This refinement involves both creating new connections and adjusting the weights of existing ones, enabling the model to better represent structural relationships.

### B.3   ADDITIONAL QUALITATIVE RESULTS

Figure 11 shows the instance adaptability of our graph prediction module. As can be seen, our graph prediction predicts different graphs for instances of the same category given different inputs. Moreover, Figure 12 shows skeleton predictions for different input values of $A_{prior}$. We use self-loops, visualizing performance without any prior knowledge, and with random omission graphs, where we drop some of the edge connections randomly. In addition, Figure 13 illustrates additional skeleton predictions.

We also present in Figure 14 additional qualitative comparison for Capeformer (Shi et al., 2023a), GraphCape (Hirschorn & Avidan, 2024), and the non graph-based SCAPE (Liang et al., 2024) on various categories.

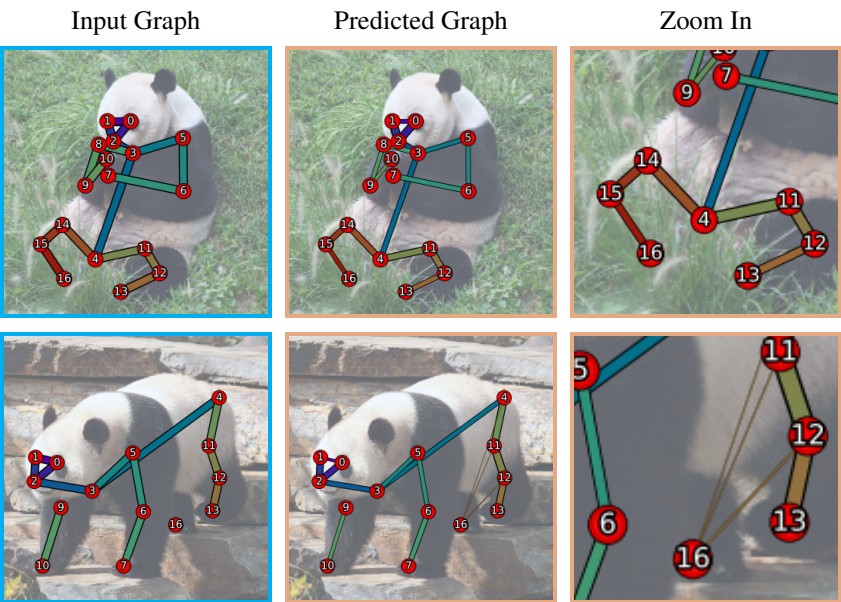

Figure 11: **Instance Adaptability.** We visualize an example of instance adaptability. The left column denotes the input $A_{prior}$ and the right column is the refined adjacency matrix. In the top row, we see a graph input of a panda body, where all keypoints are visible. In the bottom row, as some keypoints are occluded (node 15), the input graph includes isolated nodes (node 16). Our predicted graph connects this isolated node to enhance localization.

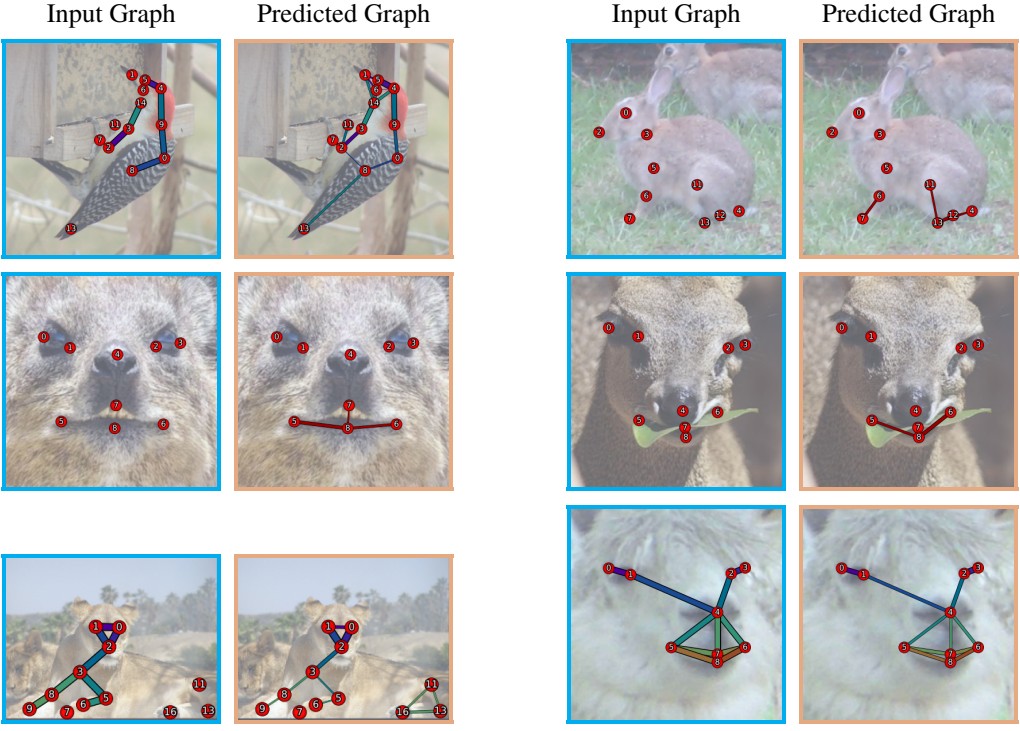

Figure 12: **Predicted Graphs with Various $A_{prior}$.** We visualize the unnormalized graph outputs with various $A_{prior}$ inputs. The left column denotes the input $A_{prior}$ and the right column is the refined adjacency matrix. Line width corresponds to edge weight.

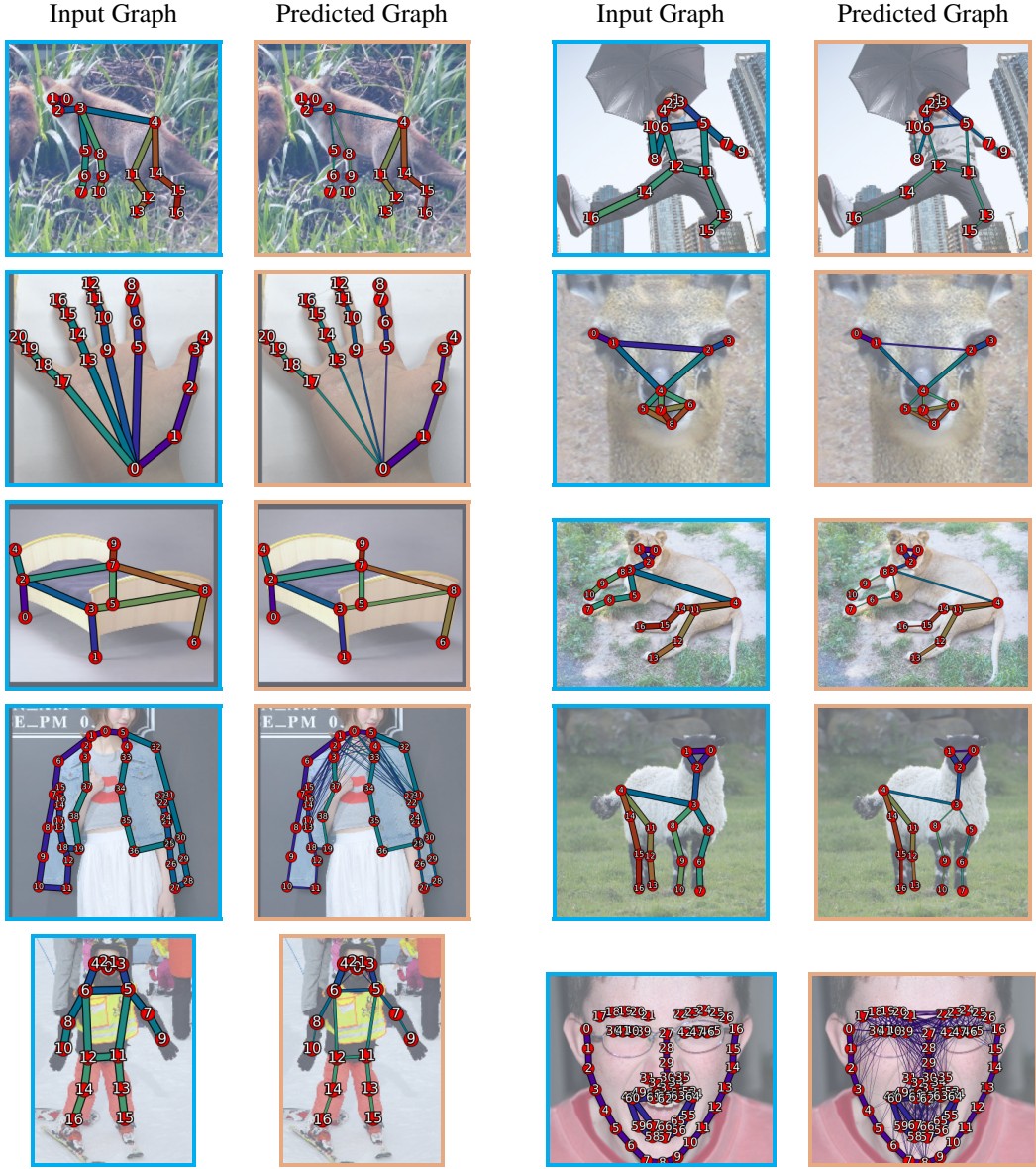

Figure 13: **Predicted Skeleton.** We visualize the unnormalized graph outputs. The left column denotes the input $A_{prior}$ and the right column is the refined adjacency matrix. Line width corresponds to edge weight. The model disconnects symmetric parts and creates new connections that are helpful for localization.

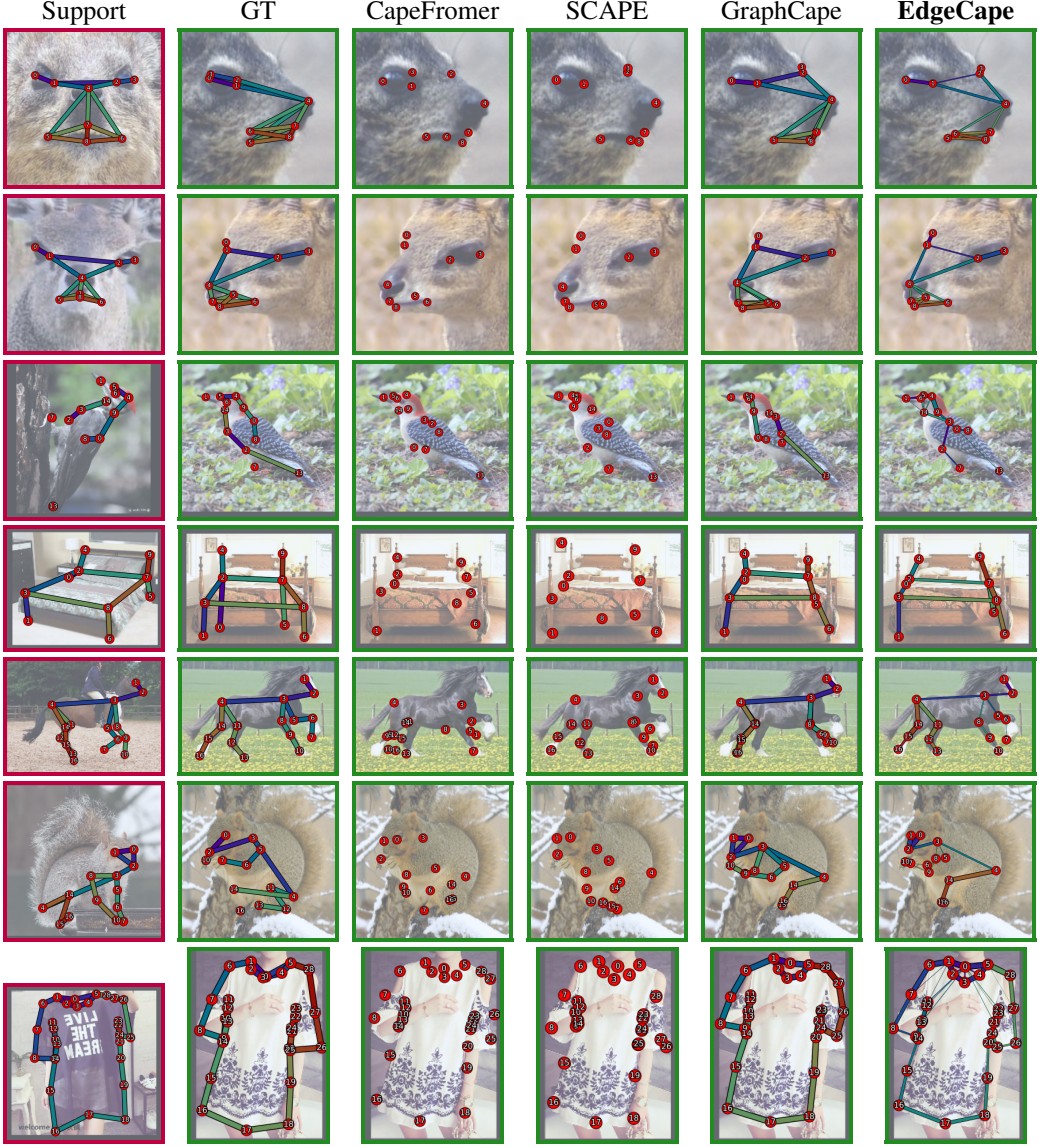

Figure 14: **Qualitative Comparison.** We visualize keypoints predictions for the 1-shot setting. The left column denotes the support image with its corresponding skeleton. The second column is the ground-truth query keypoints. The following columns are results from CapeFormer, SCAPE, GraphCape, and our method.

## C  METHOD DETAILS

### C.1  FRAMEWORK OVERVIEW

We base our method on GraphCape (Hirschorn & Avidan, 2024), introducing two main architectural modifications:

- Incorporating Markov Bias attention in the graph-decoder.
- Adding a category-agnostic pose-graph prediction module.

These modifications are detailed in the main paper, while below we provide a brief overview of GraphCape's architecture for completeness. For full details, please refer to the original paper.

- **Feature Extractor:** A pre-trained model extracts features from both support and query images. Support keypoint features are derived by element-wise multiplication between the support image's feature map and keypoint masks, created using Gaussian kernels centered at the support keypoints. In multi-shot scenarios (e.g., 5-shot), the average of support keypoint features across images in feature space is taken. This results in the query feature map $\hat{F}_q \in \mathbb{R}^{hw \times C}$ and support keypoint features $\hat{F}_s \in \mathbb{R}^{K \times C}$.

- **Transformer Encoder:** The transformer encoder fuses information between support keypoint and query patch features. It comprises three transformer blocks, each with a self-attention layer. Support keypoints and query features are concatenated before entering the self-attention layer and separated afterward. The output is the refined query feature map $F_q$ and refined support keypoint features $F_s$.

- **Similarity-Aware Proposal Generator:** GraphCape builds on CapeFormer's two-stage approach, first generating initial keypoint predictions, which are then refined via a DETR-based transformer decoder. The proposal generator aligns support keypoint features with query features, producing similarity maps from which peaks are selected as similarity-aware proposals. To enhance efficiency and adaptability, a trainable inner-product mechanism (Shi et al., 2022) is used to explicitly model similarity.

- **Graph Transformer Decoder:** A transformer decoder network decodes keypoint locations from the query feature map. It contains three layers, each with self-attention, cross-attention, and feed-forward blocks. In this design, GraphCape replaces the simple MLP in the transformer decoder's feed-forward network with a GCN-based module to incorporate structural priors directly into the keypoint prediction process. To prevent excessive smoothing—a common issue in deep GCNs that can blur node distinctions and degrade performance—GraphCape adds a linear layer for each node following the GCN layer:

$$F_s^k = W_{linear}\sigma_{act}(W_{adj}F_s^k \widetilde{A}_{prior} + W_{self}F_s^k) \qquad (14)$$

Where $W_i$ are learnable parameters, $\sigma_{act}$ is an activation function (ReLU), and $\widetilde{A}_{prior} \in \mathbb{R}^{K \times K}$ is the symmetrically normalized adjacency matrix.

Using an iterative refinement strategy (Cai & Vasconcelos, 2018; Teed & Deng, 2020; Zhu et al., 2020), each decoder layer predicts coordinate deltas for prior predictions. This means that the updated coordinates are created as follows:

$$P^{l+1} = \sigma(\sigma^{-1}(P^l) + MLP(F_s^{l+1})) \qquad (15)$$

where $\sigma$ and $\sigma^{-1}$ are the sigmoid and its inverse. Additionally, the decoder leverages predicted coordinates to provide enhanced reference points for feature pooling from the image feature map. The keypoints' positions from the last layer are used as the final prediction.

### C.2  CATEGORY-AGNOSTIC POSE-GRAPH PREDICTION

The architecture of our pose-graph prediction network $f_\theta$ is illustrated in Figure 15. Formally, we define a learnable function $f_\theta$ to learn the residual pose-graph in the form of an adjacency matrix $\Delta A \in \mathbb{R}^{K \times K}$:

$$\Delta A = f_\theta(A_{prior}, F_s, F_s^k) \qquad (16)$$

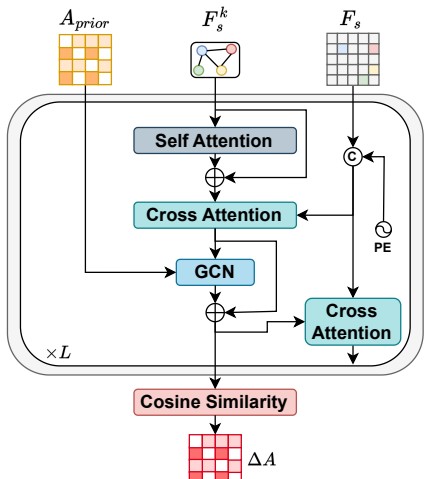

Figure 15: **Category-Agnostic Pose-Graph Predictor.** Our pose-graph predictor (as referenced in Figure 3) uses the prior input graph $A_{prior}$, the support image features $F_s$ and keypoint features $F_s^k$ to produce structure-aware keypoint features $F_{refined}^k$. Cosine-similarity is then applied to predict the residual adjacency output $\Delta A$.

where $A_{prior} \in \{0,1\}^{K \times K}$ is the unweighted graph input provided by the user (like in GraphCape), and $F_s \in \mathbb{R}^{hw \times C}$ and $F_s^k \in \mathbb{R}^{K \times C}$ are the support image and keypoint features.

To handle the few-shot setting, our pose-graph predictor refines the features of each support image independently, allowing the model to account for structure on a per-image basis. After refinement, we average the resulting structure features across all support images and predict a unified adjacency matrix. This aggregation enables the method to exploit structural cues from all available support instances, ensuring that the final structure estimate benefits from as much visible information as possible. Note that the prior skeletal relations ($A_{prior}$) are category-specific and therefore identical across all support images.

## C.3 MARKOV ATTENTION BIAS

In category-agnostic pose estimation, encoding spatial relations between keypoints is beneficial for robust localization, especially when dealing with novel objects (Hirschorn & Avidan, 2024; Rusanovsky et al., 2024). Transformer models naturally have a global receptive field, allowing each node (or token) to attend to all others within a layer. However, this flexibility introduces a challenge: Transformers lack inherent structural constraints, so positional dependencies must be explicitly encoded to reflect local relations. While this is often done in sequence data through absolute or relative positional encodings, graphs pose a different challenge, as nodes are not arranged linearly and connectivity is defined by edges.

Our approach aims to better utilize the structural dependencies between keypoints for CAPE. We build on the foundations set by GraphCape (Hirschorn & Avidan, 2024), which incorporates GCN layers into the feed-forward layers to propagate structural information. However, the GCN layers used in that model were limited by their fixed, nearest-neighbor receptive field, which restricts the model's ability to capture more complex or distant connections between keypoints.

Thus, we further integrate the graph-prior into the architecture. We follow Ying *et al.*(Ying et al., 2021), adding a bias term based on graph connectivity to the self-attention mechanism in the decoder. Denote $a_{ij}$ as the $(i,j)$-element of the Query-Key product attention matrix $a$, resulting in:

$$a_{ij} = \frac{(h_i W_Q)(h_j W_K)^T}{\sqrt{d}} + b_{\phi(v_i, v_j)} \tag{17}$$

where $b_{\phi(v_i, v_j)}$ is a learnable scalar indexed by $\phi(v_i, v_j)$, and is unique for each attention head. Unlike the limited receptive field in GCNs, using $b_{\phi(v_i, v_j)}$ enables each node in a single Transformer

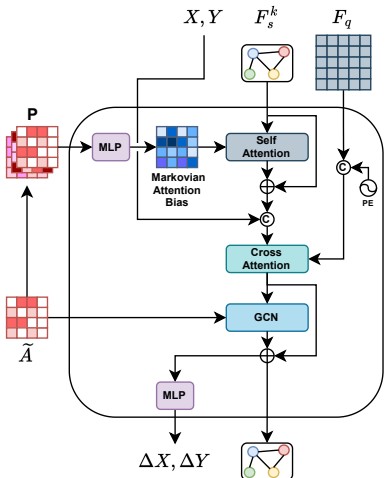

Figure 16: **Graph Decoder Prediction Layer.** Overview of the Transformer decoder architecture, adapted from the GraphCape design. The decoder consists of self-attention, cross-attention, and a graph-based feed-forward network. We incorporate a Markov Attention Bias into the self-attention mechanism to encourage structural keypoint interactions. Self-attention facilitates adaptive interactions among support keypoints, while cross-attention extracts localization information from the input features. Finally, the decoder refines keypoint features and outputs location predictions.

layer to adaptively attend to all other nodes based on the graph's structural information. $\phi(v_i, v_j)$ is usually the distance of the shortest path (SPD) between $v_i$ and $v_j$ if the two nodes are connected.

This bias term is highly effective in capturing general structure, boosting the performance of Graph-Cape by around **0.5%**. However, it assumes a discrete distance between nodes and is applicable for unweighted adjacency matrices. Our primary objective is to predict real-valued edges, representing the strength of the structural keypoints connections. Thus, inspired by Ma *et al.*(Ma et al., 2023a), we treat the normalized adjacency matrix $\widetilde{A}$ as a right stochastic matrix.

A stochastic matrix is commonly used to describe the transitions in a Markov chain, where each element $(\widetilde{A})_{ij}$ represents the probability of moving from one state (or node) $v_i$ to another state $v_j$ in a single step. Specifically, the entry $(\widetilde{A}^k)_{ij}$ gives the probability of transitioning from node $v_i$ to node $v_j$ in exactly $k$-hops. This process allows us to capture both direct and indirect relations between keypoints, enabling the model to consider more distant keypoints that may influence localization. Thus, we build the following matrix:

$$P_{ij} = [I, A, A^2, ..., A^{k-1}]_{i,j} \in \mathbb{R}^K \tag{18}$$

where $I$ is the identity matrix and the parameter $K$ controls the maximum number of hops considered. We incorporate this graph characteristic into our model's attention mechanism to enable more nuanced and flexible structural priors. The complete decoder layer is illustrated in Figure 16. Based on Equations 17 and 18, we use the following bias term:

$$a_{ij} = \frac{(h_i W_Q)(h_j W_K)^T}{\sqrt{d}} + MLP(P_{ij}) \tag{19}$$

where MLP: $\mathbb{R}^K \to \mathbb{R}$, modulates the influence of keypoints based on their distance in the graph (i.e., the number of hops between them). This formulation results in a continuous and learnable structure-based bias term.

### C.4 TRAINING SCHEME

When we attempt to directly integrate the adjacency matrix predictor and the self-attention bias mechanism, we observe only a small impact on the model's performance. Changing the adjacency

matrix while training the bias attention MLP results in unstable training. Thus, we first train our base model and fine-tune each added component.

We begin by training the base model and establishing strong foundational features necessary for robust localization. Optimization is done using $\mathcal{L}_{offset}$ which is the $L_1$ localization loss as in (Shi et al., 2023b). Once the model has converged, we integrate the skeleton predictor. This stage allows the model to further adapt by incorporating specific structural insights provided by the skeleton predictor. For this phase, we add the adjacency loss, resulting in:

$$\mathcal{L} = \mathcal{L}_{offset} + \lambda_{adj}\mathcal{L}_{adj}, \tag{20}$$

In the final phase, we maintain the frozen feature extractor and freeze the skeleton predictor. Then, we integrate the Markov bias attention. This final stage allows the model to strengthen its capacity to interpret spatial dependencies between keypoints.

This three-phase approach allows each component to integrate structural encoding progressively, enhancing accuracy through a stable framework.

## C.5 IMPLEMENTATION DETAILS

For a fair comparison, training parameters, data augmentations, and data pre-processing are kept the same as in previous works. In addition, the backbone size was matched to other works, thus, we use the smallest ($\sim$20M parameters) versions of SwinV2 and DinoV2.

The graph predicting network, encoder, and keypoint prediction decoder have 3 layers. For the Markov Bias Attention, we use a maximum of $K = 4$ hops, and for $\mathcal{L}_{adj}$ we mask 50% of keypoints, using $\lambda_{adj} = 1$. The model is built upon MMPose framework (Contributors, 2020), trained using Adam optimizer with a batch size of 16, the learning rate is $10^{-5}$, and decays by 10× on the 160th and 180th epoch. Each phase is trained for 100 epochs. Training takes $\sim$ 8 hours on a single Nvidia A100 GPU.

Evaluation of SCAPE (Liang et al., 2024) and FMMP (Chen et al., 2025a) was done by removing the keypoint identifiers (which were shown to be inapplicable for real-world scenarios (Nguyen et al., 2024; Yang et al., 2024; Hirschorn & Avidan, 2024)) and training a network using their official code. As FMMP leverages a larger number of keypoints, it experienced a relatively greater performance drop without these identifiers. For SCAPE, we use DinoV2 as the backbone to ensure a fair comparison. And, since FMMP relies on multi-scale features, we adopt a modern pre-trained multi-scale vision transformer backbone (PVT-V2). We also used GraphCape and CapeFormer official code for evaluation and visualizations.

## D  THE USE OF LARGE LANGUAGE MODELS (LLMS)

This article was refined with the help of LLMs to improve clarity and style. They helped polish the writing.

