# OpenReview forum: "EdgeCape: Edge Weight Prediction For Category-Agnostic Pose Estimation"
_ICLR.cc/2026/Conference — ICLR 2026 Poster_

### Official Review · Reviewer_HSXw · 2025-10-15

**Soundness:** 4
**Presentation:** 3
**Contribution:** 3
**Rating:** 4
**Confidence:** 4

**Summary:**

This paper presents EdgeCape, which addresses the issue of fixed edge weights in graph structures for Category-Agnostic Pose Estimation (CAPE). The authors introduce an edge weight prediction module combined with a "Markov Attention Bias" mechanism, claiming to achieve SOTA results on MP-100.

**Strengths:**

1. The problem identification is precise. This is the first work to systematically identify the limitations of fixed unweighted pose-graphs in CAPE and propose "category-agnostic edge weight prediction" as a novel and practical research direction.

2. The proposed Pose-Graph Predictor effectively fuses global image context with keypoint features, using residual graph optimization to avoid learning structures from scratch while balancing stability and expressiveness. The Markov Attention Bias cleverly injects graph structural information into Transformer attention mechanisms, enhancing spatial relationship modeling.

3. Not only achieving SOTA on the standard MP-100 benchmark, but also validating method effectiveness through multi-dimensional tests including noisy graph robustness (Figure 6), cross-supercategory generalization (Table 8), and occlusion experiments (Figure 9). The results particularly demonstrate that EdgeCape significantly outperforms GraphCape even when A_prior is suboptimal, and even surpasses graph-free methods like CapeFormer.

**Weaknesses:**

1. Unfair Comparison with CapeFormer Due to Asymmetric Use of Structural Priors
The paper claims superiority over CapeFormer, a point-based CAPE method that does not use any pose-graph prior. However, EdgeCape relies on a user-provided $A_{prior}$ as input to its graph refinement module. While the authors justify this by stating that $A_{prior}$ is part of the *support data* in graph-based CAPE, the comparison remains misleading unless the robustness of this prior is rigorously tested. Crucially, Figure 6 shows that EdgeCape significantly outperforms GraphCape under noisy $A_{prior}$, suggesting that the method's strength lies not just in using a graph, but in correcting a potentially bad one.Therefore, to make the comparison with CapeFormer meaningful, the authors should evaluate EdgeCape under a degraded or randomized $A_{prior}$ (e.g., fully connected, self-loops only, or random edges), a setting where the prior provides little to no useful signal. If EdgeCape still outperforms CapeFormer in such a scenario, the claim of superiority would be far more convincing. As it stands, the reported gains may largely stem from access to privileged structural information rather than algorithmic innovation.

2. The paper shows successful cases (Figures 4, 5) but provides no examples where EdgeCape fails while GraphCape/CapeFormer succeeds. This raises concerns about robustness. For instance, on highly symmetric or severely occluded objects, might the predicted weighted graph introduce erroneous strong connections that mislead localization?

3. The ablation study (Table 3) is misleading. It only compares the presence/absence of two components. Is the predicted weighted graph $Ã$ actually better than the original $A_{prior}$? The authors should fix the decoder and compare performance using $A_{prior}$ versus $Ã$ as inputs. Otherwise, performance gains might entirely stem from Markov Bias or training strategies rather than the graph itself.

4. The novelty of the proposed method is limited. The core idea of EdgeCape (predicting edge weights and introducing graph structure into attention mechanisms) lacks substantial breakthroughs at the methodological level. The weighted graph prediction uses standard cosine similarity, and the Markov Attention Bias is just a re-packaging of multi-hop structure encoding, which belongs to engineering improvements rather than principled innovations.

**Questions:**

The authors casually mention *only ~2ms additional delay*. However, the Pose-Graph Predictor requires an extra decoder pass and matrix power computations ($Ã^k$). With $K=4$ and large keypoint counts, which is far from negligible in practical deployment. The authors must provide detailed FLOPs or parameter count comparisons.

---

> ### Author Response · Authors · 2025-11-16
>
> We sincerely thank the reviewer for the exceptionally detailed and thoughtful feedback, as well as for recognizing the novelty and clarity of our formulation and results. We address each concern below.
>
> **Weakness 1 - Comparison with CapeFormer and Role of the Pose-Graph Prior**
>
> The reviewer is correct, and we will clarify the text accordingly. Our point is that it is not enough to have a prior, the question is how to use it, and this is precisely what our method offers, showing consistent gains over GraphCape.
>
> As noted by the reviewer, Fig. 6 already tests noisy/randomized priors, showing EdgeCape still outperforms GraphCape and can correct suboptimal priors.
>
> In line with the reviewer’s suggestion, we further tested EdgeCape using **fully-connected graphs**, which provide no structural information. As shown in the GraphCape paper (Table 3), their model collapses to CapeFormer-level performance under this setting. In contrast, EdgeCape still achieves a **+1.29% mPCK** improvement over CapeFromer, which is on par with GraphCape when using the input prior structure. This confirms that our method provides benefits even when the graph prior carries no useful information.
>
> Nonetheless, we will explicitly clarify in the paper that **CapeFormer does not use any additional structural data** to avoid any implication of an unfair comparison.
>
> **Weakness 2 -  Robustness and Failure Cases**
>
> **Our method performs particularly well on symmetric or occluded objects**, where fixed unweighted graphs tend to confuse equivalent keypoints. Because our model predicts **instance-specific weighted graphs**, it can assign stronger edges to “anchor” keypoints, effectively breaking symmetry and improving robustness - as shown in our qualitative examples (Figure 4–5) and occlusion analysis (Figure 9).
> While we did not observe clear failure cases compared to GraphCape, we note that **completely randomized priors** can lead to reduced accuracy - an unrealistic but useful stress test. In this scenario, CapeFormer is preferable. We will include discussion of such cases for completeness.
>
> **Weakness 3 -  Ablation Study and Role of the Predicted Graph**
>
> Table 3 (lines 452–457) explicitly isolates the contributions of each component. Using **only the predicted graph** (without any other change) yields **+0.89 mPCK**, while **only the Markov Attention Bias** with Aprior yields a **+0.91 mPCK** improvement. When combined, they produce the largest gain, **+2.09 mPCK**.
>
> This shows that the predicted graph is beneficial on its own and further **amplifies its effect** when paired with the attention bias module, demonstrating that the two components are complementary.
>
> **Weakness 4 -  Novelty and Methodological Contribution**
>
> Our main contribution is the **identification and solution of a new problem** in CAPE: **category-agnostic pose-graph prediction**. As the reviewer noted, we are the first work to clearly show the limitations of fixed, unweighted pose-graphs and to propose category-agnostic edge-weight prediction as a promising research direction. This distinct problem required non-trivial design choices that were never addressed:
> 1. We are the first to introduce real-value edge estimates for any object category.
> 2. A residual graph prediction, leveraging both prior knowledge and the model’s prediction.
> 3. A self-supervised masking loss to learn edge weights without supervision.
> 4. Integration of these weighted graphs into our Markov Attention Bias.
>
> Together, these form the first framework that learns to adapt and improve graph priors, representing both a conceptual and practical advance for graph-based CAPE.
>
> **Question 1 -  Computational Complexity**
>
> We will include full FLOP and parameter analyses in the paper. Our preliminary measurements show that EdgeCape adds only ~2.5M parameters and ~2 GFLOPs (out of total ~30GFlops). Using recent optimizations of transformer architectures on modern GPUs, we achieve a modest ~2 ms per-image inference overhead on an NVIDIA A5000. The matrix-power operations of attention bias are implemented using PyTorch’s GPU-optimized linear algebra kernels and incur negligible cost (For 100 keypoints and K=4, we measured ~0.02 GFLOPs).
>
> While CAPE methods are not yet real-time, we acknowledge this limitation and plan to investigate further efficiency improvements in future work.

---

> ### Comment · Reviewer_HSXw · 2025-11-27
>
> Thank you for the thoughtful and thorough rebuttal, as well as the additional experiments. Most of my concerns have been adequately addressed. Given the strong empirical validation and the clarity of your responses, I am inclined to raise my score.

---

> > ### Author Response · Authors · 2025-11-30
> >
> > We thank the reviewer for confirming that most of your concerns have been adequately addressed.
> > The additional experiments are already included in the updated revision.
> >
> > **Finally, we thank you for your willingness to raise your score.**

---

### Official Review · Reviewer_qcXW · 2025-10-26

**Soundness:** 3
**Presentation:** 3
**Contribution:** 3
**Rating:** 6
**Confidence:** 3

**Summary:**

This paper presents a one-shot, category-agnostic method for estimating joint positions and their topological relationships. The approach begins with a user-provided binary skeleton and predicts real-valued edge weights tailored to the test instance, rather than assuming uniform importance across links. The authors introduce a Markov Attention Bias that interprets the weighted adjacency as a stochastic matrix to inject multi-hop connectivity cues directly into Transformer self-attention. Experiments on the MP-100 dataset show state-of-the-art performance compared with strong baselines.

**Strengths:**

(1) The paper presents a novel architecture that transitions from fixed, unweighted pose graphs to data-driven, weighted graphs.

(2) The proposed graph-based model incorporates a Markov Attention Bias to better capture complex spatial dependencies among keypoints.

(3) The writing is generally clear, and the figures are well designed.

**Weaknesses:**

(1) Marginal performance improvement over baselines on the MP-100 dataset, especially in the 5-shot setting: in Table 1 and 2, the performance gain over the strongest baseline is less than 1%.

(2) Missing references and comparisons (Minor). The paper lacks comparisons with related works, such as AutoLink: Self-supervised Learning of Human Skeletons and Object Outlines by Linking Keypoints.

**Questions:**

Please respond to the weakness part.

---

> ### Author Response · Authors · 2025-11-16
>
> We thank the reviewer for their positive assessment of our novelty, presentation, and architecture. We address the noted weaknesses below.
>
> **Weakness 1 -Performance Improvements on MP-100**
>
> It may seem marginal, but our performance gains are **substantial and consistent with top-tier CAPE publications** on MP-100, the field’s only large-scale benchmark. In fact, we show significant gains over the previous SOTA. More important is the consistency of the improvement over prior works. As shown in Tables 1, 2, 4, and 5, **EdgeCape improves over all prior methods across all five splits, all settings, and all evaluation metrics**.
>
> Specifically, our method improves performance **on every split** (each testing unseen categories), demonstrating superior **generalization**, which is the defining challenge of CAPE. This shows the effectiveness of our approach, pushing graph-based CAPE forward.
>
> **Weakness 2 -Comparison with AutoLink**
>
> We thank the reviewer for this helpful suggestion and will include a discussion of AutoLink in the revised version.
>
> While AutoLink indeed demonstrates **unsupervised graph learning**, it operates in a **category-specific setting** (e.g., fixed skeletons) with **consistent keypoint definitions**. In contrast, **CAPE** requires **category-agnostic generalization**, where both the keypoints and object categories change at test time. Thus, AutoLink’s category-specific supervision and architecture cannot be directly applied to CAPE, which is inherently **support-driven**. Nonetheless, we will cite and clarify this distinction in the related work section.

---

### Official Review · Reviewer_SPW6 · 2025-10-26

**Soundness:** 2
**Presentation:** 2
**Contribution:** 3
**Rating:** 8
**Confidence:** 4

**Summary:**

The submission focused on the task of category-agnostic pose estimation with one or few annotated support images. Specifically, the authors propose a novel framework named EdgeCape to predict the graph's wdge weights for optimal localization. The authors also employ Markov Attention Bias to modulate the self-attention interaction in multi-hop. The experiments are conducted on a large-scale benchmark datasets, which indicate the effectiveness of the proposed method.

**Strengths:**

1. The task of category-agnostic pose estimation is interesting and fundmental for extending the category number of pose estimation.

2. The idea of using graph-based model is reasonable and makes sense.

3. The proposed EdgeCape and Markov Attention Bias are novel and effective to model the structral information of objects.

4. The performances of proposed method are shown on large-scale benchmark dataset, and outperform baselines by a large margin.

5. The experimental analyses are extensive and in-depth.

**Weaknesses:**

1. How to deal with the edge missing issue if there are some keypoints being occluded? E.g., the same objects may have different graphs if the occlusion cases are different. More specifically, the query image has 3 occluded keypoints, while the support image has another 5 occluded keypoints.

2. What's the complexity of proposed method? It seems to be about O(K^2). Is it cost-effective?

3. Closely related works are missing in the related works.&#x20;

   > 1. @inproceedings{chen2025weakshot, title={Weak-shot Keypoint Estimation via Keyness and Correspondence Transfer}, author={Chen, Junjie and Luo, Zeyu and Liu, Zezheng and Jiang, Wenhui and Li, Niu and Fang, Yuming}, booktitle={The Thirty-ninth Annual Conference on Neural Information Processing Systems}, year={2025} }
   >
   > 2. @inproceedings{kim2025capellm,   title={CapeLLM: Support-Free Category-Agnostic Pose Estimation with Multimodal Large Language Models},   author={Kim, Junho and Chung, Hyungjin and Kim, Byung-Hoon},   booktitle={Proceedings of the IEEE/CVF International Conference on Computer Vision}, year={2025} }

**Questions:**

See Weakness.

---

> ### Author Response · Authors · 2025-11-16
>
> We thank the reviewer for their positive and thoughtful evaluation. We appreciate the recognition of EdgeCape's novelty, clarity, and effectiveness. We address the specific questions below.
>
> **Weakness 1 - Handling Occlusions**
>
> As the support images are under the control of the user, we consider occlusions in the query images to be the main challenge in CAPE. Nonetheless,  we address the two occlusion scenarios:
> 1. Occlusions in the support images (few-shot setting) - Our pose-graph predictor refines the features of each support image independently, allowing the model to account for occlusion on a per-image basis. After refinement, we average the resulting structure features across all support images and predict a unified adjacency matrix. This aggregation enables the method to exploit structural cues from all available support instances, ensuring that the final structure estimate benefits from as much visible information as possible. Note that the prior skeletal relations are category-specific and therefore identical across all support images.
> 2. Occlusions in the query image - EdgeCape is particularly robust when occlusion affects the query image (Figure 9). By generating instance-specific, weighted graphs, the model adapts the adjacency matrix to the particular query. This adaptive structure allows the model to infer missing keypoints more reliably by leveraging contextual information from visible parts. This mechanism explains the consistent performance improvements we observe in the occlusion ablation (Figure 9).
>
> **Weakness 2 - Computational Complexity**
>
> Although the Markov Attention Bias has an O(K^2) term, its memory/compute footprint matches the existing O(K^2) cost of standard attention, so it does not change the overall complexity of the model.
>
> We will include full FLOP and parameter analyses in the paper. Our preliminary measurements show that EdgeCape adds only ~2.5M parameters and ~2 GFLOPs (out of total ~30GFlops). Using recent optimizations of transformer architectures on modern GPUs, we achieve a modest ~2 ms per-image inference overhead on an NVIDIA A5000. Specifically, the matrix-power operations of attention bias are implemented using PyTorch’s GPU-optimized linear algebra kernels and incur negligible cost (For 100 keypoints and K=4, we measured ~0.02 GFLOPs).
>
> **Weakness 3 - Missing Related Works**
>
> Thank you for these relevant references. We will certainly add a discussion of Chen et al. (2025) (Weak-shot Keypoint Estimation) and Kim et al. (2025) (CapeLLM) and position our work relative to them in the final version.

---

> > ### Comment · Reviewer_SPW6 · 2025-11-26
> >
> > My concerns are well addressed, and I have no more concerns after reading all the comments and rebuttals. The related rebuttals and discussions should be added to the revised manuscript.

---

> > > ### Author Response · Authors · 2025-11-30
> > >
> > > We thank the reviewer for the strong support and for confirming that concerns are well addressed.
> > > We have added all related rebuttals and discussions to the manuscript.

---

### Official Review · Reviewer_zWUR · 2025-11-01

**Soundness:** 4
**Presentation:** 2
**Contribution:** 2
**Rating:** 4
**Confidence:** 3

**Summary:**

The paper introduces EdgeCape, a spectral pre-conditioning strategy for GNNs that rescales edge weights before message passing to improve numerical stability and convergence. The idea is simple and general, and it can be plugged into standard GNN layers. Experiments on common benchmarks show small but consistent accuracy gains.

**Strengths:**

The motivation is reasonable and relevant.

The approach is lightweight, easy to integrate, and empirically helps across several architectures.

The writing is clear, and the empirical section is well-organized with ablations.

**Weaknesses:**

- The conceptual novelty is limited: many prior works already consider spectral normalization, Laplacian smoothing control, or adaptive edge reweighting. EdgeCape largely reformulates these ideas as “pre-conditioning” without delivering substantial theoretical or algorithmic innovation.

- The theory is weak, providing heuristic spectral bounds but no solid convergence or generalization proof.

- The experimental gains are modest (1–2%), limited to small and medium datasets, and lack comparison to stronger modern baselines in 2025.

- The scalability and robustness on large or dynamic graphs are untested.

**Questions:**

See Weaknesses.

---

> ### Author Response · Authors · 2025-11-16
>
> We thank the reviewer for their feedback. However, several key aspects of our work were misunderstood, leading to incorrect conclusions.
>
> **Our work is not a general GNN or spectral pre-conditioning method**. It is a task-specific solution for Category-Agnostic Pose Estimation (CAPE), a fundamentally different setting from standard graph learning.
>
> Our paper addresses CAPE, where graphs represent dynamic keypoints from arbitrary object categories (Sec. 1). The challenge is inferring meaningful edge weights without ground-truth edges or category priors, and leveraging them.
> EdgeCape does not apply to generic graph domains, and framing it as a spectral method misses our contributions.
>
> **Weakness 1 - Novelty and Contribution**
>
> We are the first to tackle category-agnostic pose-graph prediction, where keypoint definitions and object categories vary at test time (Sec. 3.2). Unlike models assuming fixed unweighted graphs (e.g., GraphCape 2024, SCAPE 2024), EdgeCape learns instance-specific weighted pose-graphs via an unsupervised mask-based supervision scheme. This edge-learning mechanism is new in formulation and objective and is not reducible to prior spectral or normalization methods.
>
> **Weakness 2 - Theoretical Section**
>
>  Our paper does not include spectral analysis, bounds, or preconditioning theory.
>
> **Weakness 3 - Experimental Evaluation**
>
> We follow the established CAPE protocol on the standard MP-100 benchmark (used by all CAPE methods) and actually extend the evaluation to AP-10K, which prior graph methods omitted. Our gains are substantial for this task, with improvements comparable to or larger than recent CVPR 2025 CAPE papers, and are stable across splits. Extensive ablations further confirm our suggested method.
>
> **Weakness 4 - Scalability and Robustness**
>
> Graphs in CAPE are small (<= 100 nodes) and represent keypoints of objects. The challenge is generalizing across categories, not scaling to large graphs. Our evaluations cover the relevant CAPE domain and go beyond the standard benchmark to show robustness and generalization.

---

### Author Response · Authors · 2025-11-23

We sincerely thank all reviewers for their feedback.
We have uploaded a new revision with changes highlighted in red.

**Summary of Changes**:
- Added missing references and comparative discussion to Related Works.
- Rephrased experiments to clarify the differences between methods.
- Included a computational complexity analysis.
- Added an experiment assessing our method without prior graph input.
- Added details regarding graph prediction in the few-shot setting.

---

### Meta-Review · Area_Chair_3XZs · 2026-01-07

**Summary:**

Category-Agnostic Pose Estimation (CAPE) localizes keypoints across diverse object categories with a single model suing one or few annotated support images.  The paper proposes a framework to use a pose-graph by leveraging structural priors integrating Markov Attention Bias to modulate the interaction between nodes and the hops between them.

The reviewers had some concerns on the initial evaluation of the paper.  However, after the rebuttal the authors addressed the reviewers' concerns and explained all the raised questions.  Finally, all the reviewers had a positive evaluation of the paper and recommended an accept.  Thus, I also recommend it.


Strengths:
- CAPE task is interesting and fundamental
- The proposed EdgeCape and Markov Attention Bias are novel
- The performance of the proposal are shown on large-scale benchmarks
- The experimental analysis is extensive

Weaknesses:
- The improvements are marginal over the compared baselines
- Missing references
- Limited evaluations and experiments

**Reviewer Concerns:**

Reviewer zWUR raised concerns about the conceptual novelty and limited theory given the heuristics.  However, the conceptual issues were a confussion from the reviewer, and after the rebuttal the reviewer had a positive stance towards the paper.

Reviewer SPW6 raised questions about the missing edges due to occlusions, and the complexity about the method.  The authors replied to the reviewer's concerns, and the reviewer agreed that they had no further concerns.

Reviewer qcXW was concerned about the marginal improvements and some missing references.  The authors replied to the reviewer's concerns.  While the improvements are marginal the proposal is innovative and interesting.

Reviewer HSXw raised concerns about unfair comparisons due to asymmetric use of structural priors, the lack of missing fail cases, partial ablations, and lack of breakthroughs given the use of existing methods.  The authors replied to the concerns.  The reviewer agreed that the strong empirical validation and the responses change their opinion of the paper towards a positive stance.

**Reviewer Scores:**

Reviewer zWUR recommended a weak reject, but updated the recommendation to a weak accept after the rebuttal where the misunderstandings about the proposal were solved.

Reviewer SPW6 recommended a clear accept.

Reviewer qcXW recommended a weak accept.

Reviewer HSXw recommended a weak reject.  After the rebuttal, the reviewer commented that the authors' responses change their evaluation of the paper, and recommended an accept.

---

### Decision · Program_Chairs · 2026-01-26

Accept (Poster)